# NSC Physiological Features in Spinal Muscular Atrophy: SMN Deficiency Effects on Neurogenesis

**DOI:** 10.3390/ijms232315209

**Published:** 2022-12-02

**Authors:** Raffaella Adami, Daniele Bottai

**Affiliations:** Department of Pharmaceutical Sciences, Section of Pharmacology and Biosciences, University of Milan, Via Balzaretti 9, 20133 Milan, Italy

**Keywords:** spinal muscular atrophy, induced pluripotent stem cells, differentiation, survival motor neuron, epigenetic, mitochondria

## Abstract

While the U.S. Food and Drug Administration and the European Medicines Evaluation Agency have recently approved new drugs to treat spinal muscular atrophy 1 (SMA1) in young patients, they are mostly ineffective in older patients since many motor neurons have already been lost. Therefore, understanding nervous system (NS) physiology in SMA patients is essential. Consequently, studying neural stem cells (NSCs) from SMA patients is of significant interest in searching for new treatment targets that will enable researchers to identify new pharmacological approaches. However, studying NSCs in these patients is challenging since their isolation damages the NS, making it impossible with living patients. Nevertheless, it is possible to study NSCs from animal models or create them by differentiating induced pluripotent stem cells obtained from SMA patient peripheral tissues. On the other hand, therapeutic interventions such as NSCs transplantation could ameliorate SMA condition. This review summarizes current knowledge on the physiological properties of NSCs from animals and human cellular models with an SMA background converging on the molecular and neuronal circuit formation alterations of SMA fetuses and is not focused on the treatment of SMA. By understanding how SMA alters NSC physiology, we can identify new and promising interventions that could help support affected patients.

## 1. Neural Stem Cells

Stem cells can be classified according to their species (i.e., human or murine), derivation source (i.e., embryonic, placental from embryonic appendages, or adult tissues), and differentiation power (i.e., the number of cell types that can be created by differentiation). The zygote, derived from the fertilization of an egg cell by a sperm cell, is a totipotent stem cell (up to the eight-cell stage in humans) because it can generate all cell types in our body, including embryonic appendages. In early embryonic development stages, cells become increasingly specialized. There are embryonic stem cells (ESCs) in the inner blastocyst (cell) mass that are considered pluripotent and can differentiate into all cells of an adult organism except those constituting embryonic appendages [1,2,3,4].

Multipotent cells (e.g., hematopoietic cells capable of giving rise to all blood cell types) are created from pluripotent cells, then transform into oligopotent cells, and finally, unipotent cells that can differentiate into only one cell type. It is important to note that the differentiation process never ends. In the adult tissues, some regions continue to contain and maintain stem cells that can differentiate into various cell types of the tissue, where they mainly play a homeostatic role. They remain undifferentiated and retain the ability to differentiate into other cell types. These stem cells can not only differentiate but also self-renew.

The first nervous tissue stem cells (NSCs) are formed during the initial development stages, starting from neuroepithelial progenitors in the neural tube and creating radial glial stem cells. These cells perform a double function: as scaffolding along which post-mitotic neuroblasts migrate and as progenitors. After the nervous system’s developmental phase, most radial glial stem cells lose their ability to differentiate into neurons but retain the ability to form glia. The exceptions are radial glial NSCs found in adult neurogenic zones (niches). In recent decades, the subventricular zone (SVZ) [5,6] and hippocampal dentate gyrus [7] have been described as the two main neurogenic niches, especially in rodents. However, it is possible to identify and isolate this cell type in other brain regions in some animals [8,9]. In neurogenic niches, new neurons are created, which can repopulate particular central nervous system (CNS) districts. For example, the rodent SVZ has a homeostatic function, producing cells that replace glomerular and periglomerular neurons in their olfactory bulb [10]. While this function has not yet been clarified in humans, some studies have shown that other brain regions can generate new neurons [11,12,13]. However, this function remains controversial since other studies could not find NSCs in the adult human brain [14,15].

NSCs in the adult nervous tissue were among the last to be identified and isolated in a living organism. Their acceptance by the international scientific community was not immediate since the dogma “no new neurons” introduced in 1913 by Santiago Ramón y Cajal, in which mammalian neurons are not replaced during life, was difficult to overcome [16,17]. Stem cells can be identified in nervous tissue using two methods depending on whether research is performed on fixed tissue or cells are isolated and cultivated in vitro. While these two methods are not exactly equivalent, they are complementary in some respects. Starting from fixed tissue, we can define NSCs based on their intracellular and membrane marker expression. The immunohistochemical identification of human NSCs, particularly those in the hippocampus, involves neurofilament markers such as nestin, doublecortin, neuronal nuclei protein, β-tubulin 3, light neurofilament polypeptide, microtubule-associated proteins (e.g., MAP1b/2/5) [18,19,20,21], and cell proliferation proteins (e.g., Ki67) [20] known to be present during relatively early neuronal development stages.

The reference method consists of characterizing NSCs based on their physiological properties. The first NSC feature is their ability to self-maintain. Each NSC performs self-maintenance either through forced asymmetric division, where they generate another stem cell and a cell destined to differentiate, or stochastic differentiation, where a stem cell population is conserved by approximately equal numbers of stem cells generating other stem cells via self-replication and generating cells destined to differentiate [22,23].

Another NSC feature is their capability to differentiate into the CNS’s cellular subtypes (e.g., neurons, oligodendrocytes, and astrocytes) and to change their functional characteristics in response to their microenvironment. However, it should be noted that these cells do not behave the same in vitro as they do in living tissue [24]. This functional dichotomy is an interesting aspect: while radial glial NSCs found in the adult neurogenic zones can form only new neurons in vivo, they retain the ability to produce glia in vitro because the exposure to epidermal growth factor arrests the production of neuroblasts becoming more astrocytes-like [25].

### Induced Pluripotent Stem Cells (iPSCs) as NSC Precursors

Until recently, it was believed that ESCs isolated from the blastocyst’s inner cell mass could meet the therapeutic needs of numerous diseases since they can generate all cells in an adult organism. However, in addition to serious ethical issues relating to the destruction of embryos to obtain ESCs, serious clinical problems arose, including teratomas forming from undifferentiated transplanted ESCs. In 2006, Yamanaka [26,27] developed a very innovative method to produce pluripotent cells similar to ESCs from adult cells (initially fibroblasts). They identified 4 genes from 24 candidates as necessary and sufficient to induce pluripotency (hence the name, iPSCs): SRY-box transcription factor 2, MYC proto-oncogene BHLH transcription factor, KLF transcription factor 4, and POU class 5 homeobox 1 (Pou5f1/Oct4). Subsequent studies have made it possible to confirm that these cells behave like ESCs and have the ability to generate all cells in the adult organism [28]. They also showed that iPSCs could be produced from adult cells other than fibroblasts, such as other skin cells [29], blood cells [30,31,32,33], and cells from other adult tissues [34,35,36] or in body waste products, including urine [1].

Therefore, it is possible to obtain cells similar to ESCs from an individual’s cells and then differentiate them into any adult cell type, such as liver, muscle, or nerve. Nerve cells are of particular interest since, as we have seen, using NSCs is limited both by the isolation procedures from deceased donors and the available immunological profiles. Instead, iPSCs enable the production of neurons and other nervous tissue cells from the cells of living individuals [37,38].

## 2. Spinal Muscular Atrophy

Spinal muscular atrophy (SMA) is a very severe disease that mostly affects newborns, children, and young adults. It causes the death of the lower motor neurons (MNs) and, in its more severe forms, patient death at a very young age due to respiratory problems. SMA is an autosomal recessive disease affecting 1:6000–1:11,000 individuals. Indeed, it is the second most frequent disease affecting the young after cystic fibrosis, and carriers are 1:47 among Caucasians and 1:72 among Afro-Americans [1,39].

SMA was first described in the late 19th century by Guido Werdnig and Johann Hoffman [39,40]. It displays a wide clinical spectrum, from embryonic lethality to adult-onset, and can be divided into eight different phenotypes [1,41]. Type 0, also called 1A, is diagnosed during prenatal life since the fetus reduces its movements; soon after birth, patients readily require respiratory support and die within the first month. Types 1B and 1C, also called Werdnig–Hoffman disease, are also severe SMA forms. Their onset is before 6 months and is indicated by the patient’s inability to sit, so they remain prone and require respiratory support; if untreated, patients die early, but with respiratory support they can live for two years on average. Type 2 is an intermediate phenotype with onset between 7 and 18 months; patients can sit but cannot walk and usually live for >40 years. Types 3 and 4 are less severe forms. Type 3 has an onset after 18 months of life and can be divided into 3a, where the patient loses the ability to walk during adulthood, and 3b, where the patient retains the ability to walk; their life span is similar to the general population. Type 4 is so mild that, in many cases, it cannot be diagnosed; its onset is in the second and third decades of life, and patients have a similar walking capacity and life span to the general population [1,41,42,43,44].

SMA is caused by the inactivation of the survival MN (*SMN1*) gene located in an unstable telomeric region of chromosome 5. This inactivation can be due to point mutations or larger deletions. SMN is highly conserved in vertebrates [45]. In humans, SMN comprises 294 amino acids (aa) and multiple domains: an N-terminal Gem nuclear-organelle-associated protein 2 (GEMIN2)- and nucleic acid-binding domains, a central Tudor domain, and C-terminal proline-rich (PLP) and YG-rich domains. Alterations in all SMN domains have been associated with SMA [46], suggesting that its structure is critical for its functions in humans.

SMN performs the main role in establishing nuclear gems that share numerous constituents with nuclear Cajal (coiled) bodies (CBs) and are involved in the storage and maturation of ribonucleoprotein (RNP) complexes comprising small nuclear (snRNPs), small CB-specific, and small nucleolar RNPs and telomerase RNP complexes [45,47]. Cytosolic SMN localizes to microtubules, the Golgi network, the sarcomeric Z-discs, and cytosolic stress granules (SGs) [45,48,49,50].

Early investigations struggled to understand and explain SMA’s variable severity since the simple inactivation of both alleles could not explain it. However, the discovery of the survival of multiple motor neuron 2 (*SMN2*) centromeric gene copies clarified this issue [51,52]. *SMN2* is very similar to *SMN1* except for 11 nucleotide (nt) changes, most of which do not alter its aa sequence [53]. A C6T change in exon 7 is particularly important since it alters a splicing silencer motif without changing the aa sequence. Its presence allows for exon 7 excision and produces a truncated mRNA that is translated into an inactive protein containing an EMLA sequence encoded by exon 8 that acts as a degradation signal for *SMNΔ7* [54]. Only 10% of *SMN2* transcripts escape splicing and produce a full-length mRNA (FL-SMN) and active protein, which is insufficient for normal function. However, *SMN2* can be present in multiple copies, compensating for inactivated *SMN1* [55]. Numerous proteins have been suggested as SMA severity modifiers, including Plastin 3, influenza A virus (H4N4), neuronal apoptosis inhibitory protein, ZPR1 zinc finger protein (ZPR1), insulin-like growth factor 1 (IGF1), and ubiquitin-like modifier activating enzyme 1 [56,57,58,59,60]. Nevertheless, these factors cannot fully compensate for the loss of SMN functions.

### SMN Isoforms

SMN is a ubiquitarian protein whose absence mostly affects MNs. However, other cells in the brain and peripheral tissues are involved in more severe SMA forms. The reason why SMN protein insufficiency causes MN death remains incompletely understood. However, many studies highlight the fact that MNs are a very peculiar cell type that requires functional SMN1 more than other cells [61], likely due to its multifunctional roles.

*SMN1* and *SMN2* generate various transcripts under normal and oxidative-stress conditions [62,63,64,65].

SMN-FL and SMN∆7 are highly produced during development but decay in the adult CNS. The former is located in nuclear and cytosolic gems, including axons, dendrites, and synapses [49,66]. The latter accumulates in the nucleus [54,67].

MRNAs retaining intron 3 produce an axonal-SMN (a-SMN) protein that plays a developmental role [62]. This splice variant can stimulate axon growth and cell motility and regulates the expression of chemokines such as C-C Motif chemokine ligands 2 and 7 and IGF1 [68]. In addition, a-SMN and FL-SMN are necessary for neuronal polarization and axon and dendrite organization [69]. The degradation of *a-SMN* transcripts in adult tissues is possible via the nonsense-mediated decay pathway. Other splice variants, such as those generated by the exonization of an Alu-like sequence present in intron 6, are more stable than *SMNΔ7* [65]. However, similar to SMN isoforms produced by skipping exons 3 or 5, their function remains unknown [65,70].

About 10%–50% of *SMN2* pre-mRNA is properly spliced and produces functional SMN depending on the tissue [71]. Exons 2A and 2B are conserved and are involved in key protein functions such as nucleic acid binding, signal recognition particle biogenesis, DNA recombination, snRNP assembly, and translation regulation [45,72]. This region also interacts with p53, a tumor suppressor protein and transcriptional regulator [72]. Exon 3 encodes a Tudor domain involved in interactions with proteins with RGG/RG motifs [73,74,75] such as fragile X mental retardation protein; GAR1 RNP; Ewing’s sarcoma protein; fibrillarin; fused in sarcoma; heterogeneous nuclear RNPs Q, R, and U; histone 3; and Sm proteins [45]. A proline-rich sequence involved in the actin dynamics is present downstream of the Tudor domain. This region comprises the last sixteen aa encoded by exon 7 and a YG box encoded by exon 6, enabling self-oligomerization [45].

SMN’s C-terminal sequences, including the YG box, interact with ZPR1, transcription co-repressor SIN3 transcription regulator family member A, and dead-box helicase 20 (also called gemin-3) [76,77,78]. Truncated SMN2, which represents 90% of the total protein produced by *SMN2* in the absence of functional *SMN1*, cannot interact with enzymes such as trimethylguanosine synthase 1, small nucleolar RNAs, and an mRNA subpopulation [79,80]. It also lacks the QNQKE signal sequence required for cytoplasmic localization and axonal function [81].

In addition, SMN translationally regulates protein arginine methyltransferase 4, a multifunctional protein affecting transcription, splicing, and autophagy [82,83,84]. SMN downregulation increases coactivator-associated arginine methyltransferase 1 [85], and selenoprotein synthesis levels. However, the consequences of low SMN levels on the synthesis of various selenoproteins in different tissues remain to be explored. SMN in SGs is also involved in DNA recombination and repair since the RAD51 recombinase’s interaction with the SMN-GEMIN2 complex enhances RAD51-mediated homologous pairing and strand exchange reaction in vitro [86,87]. Furthermore, SMN is involved in signal transduction pathways, such as those regulating neurites and growth cones. Finally, SMN might have a protective role in MNs, specifically in growth cones [88], intracellular trafficking, endocytosis, and autophagy.

## 3. Physiological and Molecular Differences in Healthy and SMA NSCs

### 3.1. Epigenetic

Epigenetic DNA mechanisms include the frequent methylation of cytosine residues’ five positions, inhibiting neighboring gene expression. Common histone modifications include acetylation, ubiquitination, and methylation of lysine or arginine residues in the histone tails, either promoting or repressing gene expression.

While DNA methylation patterns are relatively stable in terminally differentiated cells, they are strikingly diverse among different tissue stem cell types and change dynamically during development. Whether methylation levels among *SMN2* copies influence NSC proliferation or differentiation remains unclear. Among the various DNA (cytosine-5)-methyltransferase (Dnmt) families [89]), the Dnmt3 family contains at least two essential members with nonoverlapping functions in development: Dnmt3a and Dnmt3b. Their inactivation eliminated de novo methylation in mouse embryos, causing death four weeks after birth (Dnmt3a) or embryonic lethality (Dnmt3b) [90], and increasing NSC proliferation.

DNMT3a and DNMT3b catalyze the de novo addition of methyl groups to DNA, while DNMT1 maintains methylation patterns in newly synthesized DNA [91]. Dnmt3 orthologs have a variable N-terminal region (~280 aa in Dnmt3a and ~220 aa in Dnmt3b) in many species, including humans and mice. Nevertheless, they all contain a PWWP motif [92]. Detailed PWWP sequence analysis indicated that this domain comprises a five-stranded β-barrel structure similar to the SMN Tudor domain [93]. However, the structural similarity between Tudor and PWWP domains remains uncertain.

Non-CG DNA methylation (mCH; where H = A, C, or T) is enriched in mouse and human neurons compared with other cell types. It occurs primarily at cytosines preceding an adenine (mCA). The mCH rate is lower than the mCG rate. Conversely, in some neuron classes, the number of modified CH sites surpasses those of modified CG sites [94].

Dnmt3a catalyzes the de novo modification of mCG and mCH sites. Its expression is upregulated in neurons starting at birth and peaking at 2 weeks before plateauing by 4–6 weeks in the frontal cortex and declining in adulthood in the mouse [95,96]. Full mCH accumulation takes 16 years in humans, even though most sites are formed in the first 2 years. Furthermore, it has a heterogeneous distribution with no mCH deposition at completely silent genes or inaccessible constitutive heterochromatin regions but appreciable accumulation at extragenic regions, repeated sequences, inactive regulatory elements, and lowly transcribed genes. However, DNMT3A binding and mCH accumulation are missing from highly expressed genes and active regulatory elements [97].

At a local (kilobase) scale, mCH exhaustion at genes and regulatory elements mostly aligns with mCG patterns. Nevertheless, mCH shows distinctive large (megabase) scale patterns most likely associated with chromosome folding within the nucleus and topologically associating domains in chromatin folding [98].

The association between gene transcription and chromatin folding is undoubtedly associated with mCH deposition. While the mechanisms regulating Dnmt3a remain unclear, the analysis of different histone modifications in mouse cortex indicates that mCH deposition is controlled by chromatin structure, especially during early postnatal development [99].

Non-CG methylation is known to be highly cell-type specific, either at local or global levels. For example, in mice and humans, mCH levels can vary by up to 2-fold between brain regions [94,95,96], 1.5-fold among neuron subtypes present in the same brain region [100,101], and increases in cortical excitatory neurons from the upper to deeper layers [102].

*DNMT3a* expression decreased through various differentiation phases (i.e., dorsal fate specification, neural progenitor cells [NPCs], self-organized rosettes, and maturing neuronal cells) in an iPSC corticogenesis model [103]. Presumably, the methylation level of various genes also decreases within those differentiation phases.

*SMN2* is also subject to silencing by DNA methylation. Indeed, *SMN2* contains four CpG islands with conserved methylation patterns. They contain 85 CpG dinucleotides, 14 of which are between nts −896 and −645, 12 between nts −469 and −247, 38 between nts −151 and 296, and 21 between nts 844 and 1146.

Fibroblast cell lines from SMA patients showed almost complete methylation at CpG1 and CpG4, while CpG2 was partially methylated and CpG3 showed very little methylation [104]. CpG methylation (at positions −290 and −296) in blood-borne cells correlated with disease severity and first transcriptional start site activity at *SMN2* at position −296 in patients affected by severe SMA and suffering from mild SMA but carrying identical *SMN2* copy numbers [104]. Inhibiting *SMN2* silencing via DNA methylation is considered a promising pharmacologic SMA therapy [104,105].

*SMN2* expression does not change significantly during healthy cell development and differentiation. This observation is expected since *SMN1* is the primary SMN gene expressed in healthy individuals. However, *SMN1* expression showed the same pattern [103].

Nevertheless, a deeper analysis showed some differences in SMA neural development. *SMN2* showed hypomethylation in early development (iPSCs, neuroepithelial precursors (NEP), and immature (IMN) and mature (MMN) MNs) that most likely reflects its increased expression in cells derived from SMA1 patients than from SMA2 patients or healthy individuals to compensate for *SMN1*′s absence (Figure 1) [106].

More specifically, the methylation at CpG2 varied from stage to stage with no clear tendency (Figure 1). However, while CpG4 methylation in SMA1 patient-derived cells was significantly lower at most development stages, it was significantly lower in the final two MN differentiation stages (IMM and MMN) in SMA2 patient-derived cells (Figure 1B).

The same study [106] compared the DNA methylation of genes involved in differentiation from iPSCs to NEP to MNs, starting from genes involved in maintaining the pluripotency state particularly relevant to iPSCs. *OCT4* is commonly hypomethylated in ESCs or iPSCs [107,108]. However, it showed greater methylation in iPSCs from SMA1 and SMA2 patients than from healthy individuals. In contrast, the lysosomal trafficking regulator gene is hypermethylated in these cells [109] and iPSCs from SMA1 patients and healthy individuals. However, its methylation was significantly higher in iPSCs from SMA2 patients than from SMA1 patients and healthy individuals (Figure 2A).

These findings indicate that methylation plays an important role in maintaining iPSC pluripotency. Remarkably, significant increases in *OCT4* methylation levels in SMA cells and spalt-like transcription factor 4 gene methylation levels in SMA1 patient-derived cells were present in the three final MN differentiation stages (MN precursor, IMN, and MMN).

Methylation analysis of genes involved in neural differentiation and functioning indicated that primary pathways involved in MN differentiation and maturation are heavily affected in SMA. For example, methylation levels of the MN and pancreas homeobox 1 gene activated during MN maturation [110] were inversely correlated with its expression level [111]. Its methylation levels were also significantly increased in IMNs and MMNs reprogrammed from SMA1 patient cells, most likely reducing transcription efficiency (Figure 2C,D).

Choline acetyltransferase (CHAT) is a terminal MN marker expressed in MMNs [110]. It showed significantly increased methylation in iPSC-derived MNs in cell cultures from a severe SMA case that might affect its expression [106] (Figure 2D). While low CHAT levels are correlated with the loss of MN function, decreased CHAT activity in SMA MNs is not the main factor causing their degeneration [71]. Instead, acetylcholine receptor clustering changes were described in early SMA1 development [112]. Nevertheless, increased CHAT levels might be important in ameliorating MN pathology [113].

Plastin 3 (PLS3) and neurocalcin delta (NCALD) are of particular interest since there are many reports assessing their role as SMA modifiers [114,115]. Indeed, PLS3 acts as a positive disease modifier via endocytosis regulation and is capable of binding to actin and regulating cytoskeletal dynamics through various mechanisms, including actin filament bundling [88,116]. NCALD adversely affects endocytosis and SMA severity. However, methylation analysis was inconclusive due to the low number of SMA specimens [106]. Nevertheless, paired box 6 (*Pax6*) expression initiates at embryonic day 8 in mice and is present in the neural plate formed by proliferating neuroepithelial cells. *Pax6* expression is localized to the spinal cord, forebrain, and hindbrain during regionalized neural tube organization two days later. After birth, *Pax6* is strongly expressed in neurons in the amygdala, cerebellum, thalamus, and olfactory bulb. However, it is only modestly expressed in the hippocampus’ subgranular zone (SGZ) and the SVZ [117].

In the SGZ, *Pax6* is expressed in the SGZ’s neural stem and early progenitor cells. In the SVZ, it persists in migrating immature and mature neurons and is required for the specification of dopaminergic periglomerular cells and GABAergic granule cells. In vitro experiments have shown that PAX6 is more abundant in early differentiation phases, specifically in dorsal fate specification, which is an active proliferating stage [103]. Postnatally, the downstream gene, fatty-acid-binding protein 7 (*FABP7*), has a pivotal role in maintaining neural stem and progenitor cell proliferation during hippocampal neurogenesis. Indeed, in vitro studies also indicated that *FABP7* expression is higher in NPCs [103].

*PAX6* methylation is consistently higher in all differentiation stages in SMA-derived cells. In particular, *PAX6* expression increases neurogenesis by human striatal NSCs [118]. Therefore, increased methylation and consequent inactivation of *PAX6* in NEPs and also iPSCs (Figure 2A) and IMNs and MMNs (Figure 2C,D) [106] is associated with reduced stemness in SMA-derived NSCs. Significantly increased *PAX6* promoter methylation in cells from SMA1 and SMA2 patients at each consecutive neural differentiation stage might be responsible for changes in *PAX6* expression and disrupt the maturation of MNs already damaged in the SMA condition [106] (Figure 2).

The methyl CpG-binding protein 2 (MECP2) is disrupted in the Rett syndrome and is a major mCH methyl marker reader. Indeed, it is clear that MECP2 accumulates in neurons throughout postnatal development in parallel with mCH [119] reaching an expression level comparable with that of histone H4 [120], *MECP2* expression in neurons is essential for nervous system function [121], and does not change during iPSC differentiation [103].

A very recent study found a relationship between MECP2 and SMN2 [122]. Indeed, MECP2 binds the *SMN2* promoter at nts −167 to −43, and when this site is blocked by antisense oligonucleotides, at −372 to −43, *SMN2* expression is additively enhanced with the drug nusinersen [122]. This result also translates to an FL-SMN protein increase in patient fibroblasts, an increase in the transactional quadricep and intercostal muscle areas of the mouse model ∆7 (FVB.Cg-Grm7^Tg(*SMN2*)89Ahmb^
*Smn1*^tm1Msd^ Tg(*SMN2* ∗ delta7) 4299Ahmb/J), and increased survival [122].

Conversely, the Tudor domain is involved in the methylation reading of proteins, particularly histones. Indeed, the Tudor domain of Tudor-domain-containing protein 3 recognizes arginine demethylation with asymmetric H4R3, H3R17, and H3R2 dimethylation. It was discovered over two decades ago that SMN binds dimethylated glycine and arginine-rich motifs of snRNP Sm’s D1 and D3 via its Tudor domain [123,124].

### 3.2. SMN Protein Dosage and NSCs Properties

A milestone study [125], produced NSCs from embryonic day 14.5 mouse embryo striata from a severe SMA model with Smn1/2 genotypes. These cells showed proliferation and clonal capabilities similar to those obtained from wildtype animals. However, they produced fewer Tuj1-positive neuronal cells, with fewer and shorter neurites. Interestingly, the reduction in Tuj1-positive cells was related to an increase in nestin-positive cells, indicating that these cells have some differentiation impairment.

The reduction in neurite numbers and lengths in SMA-derived neuronal cells can be directly related to alterations in various pathways in which SMN is involved. For example, profilin proteins can bind concurrently to actin (Figure 2C,D) and proteins containing the PLP domain [126] which is also present in SMN and encoded by exon 5 [127]. In mammals, the 12–15 kDa profilin proteins are encoded by four genes. Profilin-1 is ubiquitously expressed, while profilin-2′s most abundant splice variant, isoform 2a, is predominantly present in the nervous system [128]. Profilins exert two main effects on F-actin formation. They bind to actin monomers and ADP-ATP exchange for G-actin, inhibiting F-actin formation [129]. Moreover, profilin-2 has a greater affinity for PLP domains than profilin-1 [130]. Indeed, the SMN-profilin-2 interaction is more noticeable than the SMN-profilin-1 interaction. Profilin-2′s primary expression in nervous tissue explains why SMA shows neuron degeneration but no actin dynamic alteration in other cells [56,127] (Figure 2C,D). Rho-associated kinase (ROCK) regulates profilin-2a via phosphorylation [131] resulting in its hyperphosphorylation. This mechanism contrasts with other ROCK targets, such as cofilin, which are hypophosphorylated (Figure 2D) [127].

Reduced neurite length was also found using iPSCs from healthy individuals or SMA1 or SMA3 patients [132]. After iPSCs creation, these cells were differentiated into NSCs and then MNs. Interestingly, the average total neurite length of both SMA clones and neurite projection from SMA3 clusters were much longer than that of the SMA1 clusters [132].

Other studies using different technical approaches [133] have shown that MNs differentiated from SMA patient-derived iPSCs have altered neurite growth. Indeed, iPSCs displayed behavior that appears to change depending on the SMN protein during the various differentiation stages.

An interesting study by Altelaar et al. [134] showed that during early iPSC differentiation phases, such as neuralization, differences in SMN levels between SMA and controls started to increase from T5−T7 onwards when dendrites and axons begin to develop and mature. However, Varderidou-Minasian et al. [134] showed this difference is limited at the neuralization stage (T2, T3, and T4), suggesting that differences in SMN protein levels are minimal at the NSC stage. It is possible to hypothesize that SMN protein levels are not so high in the early development stages. Therefore, the amount produced by *SMN2* is sufficient to support this development phase. However, the higher SMN levels required during MN differentiation and maturation to support appropriate development cannot be met by *SMN2* alone.

Wildtype animals had lower *Smn1* expression in the thalamic region than in the hippocampus. Similarly, this region’s cell density, morphology, and proliferation are strongly affected in SMA animals. Indeed, morphological assessment of an *Smn2*/2;*SMN2* mouse model of severe SMA at pre- and late-symptomatic time points found hippocampal-specific effects from reduced SMN levels. In particular, the hippocampal dentate gyrus was noticeably smaller in *Smn2*/2;*SMN2* mice at late symptomatic time points [135].

### 3.3. SMA, Mitochondria and NSCs

Unlike mature neurons, NSCs use glycolysis and the pentose phosphate pathway for their cellular metabolism, producing energy (ATP) and metabolites (pyruvate and nicotinamide adenine dinucleotide phosphate) that are essential building blocks for aa and nt synthesis to support their high cell division rates [136].

NSCs derived from mouse cortex [137,138], human ESCs (hESCs) [139], adult NPCs derived from human iPSCs [140], and adult mouse NPCs [141].

NSC proliferation and neuronal cell differentiation are accompanied by morphological or functional modifications at the mitochondrial level. These changes are true during embryonic and adult neurogenesis and NSC proliferation and differentiation into NPCs that differentiate into mature neurons. Since mature neurons need more energy to sustain homeostasis and support their specialized functions than stem cells, switching from glycolysis to oxidative phosphorylation is necessary during neuronal differentiation [142].

These changes follow a progression in mitochondrial morphology, varying from fragmented to elongated throughout the differentiation stages. The currently accepted concept is that before differentiation, mitochondria are poorly functional with a fragmented morphology and immature cristae structure in early development phases [143,144,145] or different tissues’ adult stem cells [146]. They become activated during differentiation through mitochondrial elongation and increased cristae number [147]. This aspect was also shown in NSCs during embryonic and adult neurogenesis, with a change in mitochondrial morphology as NSCs began to differentiate first to a neuronal fate, as NPCs, and then into neurons [142].

It is interesting to consider the effects of SMN deficiency or absence in the mitochondria of SMA NSCs in the context of these hypotheses. Contrasting with earlier findings [148,149], it appears that SMN directly or indirectly interacts with many mitochondrial proteins. SMN directly interacts with mitochondrial proteins such as translocases located on the inner membrane, such as translocase of inner mitochondrial membrane 50 [150], p53, and B cell lymphoma-2. SMN also has indirect interactions via SMN-interacting proteins that localize to mitochondria, such as NCALD and stasimon, which interact with voltage-dependent anion-selective channel 1 (VDAC1) between the endoplasmic reticulum and mitochondrial membranes [151] (Figure 2D). The VDAC1-stasimon interaction could be disrupted by reduced SMN levels, this effect links mitochondrial protein import with SMA pathology [152] (Figure 2D).

These aspects correlate with mitochondrial morphology and size. However, other cell types are also associated with the mitochondrion’s energetic and metabolic output. Mitochondrial morphological abnormalities are present in animal SMA models, such as alterations in cristae and larger and smaller sizes [153] (Figure 2C,D).

Additionally, the knockdown of *Smn* in a cellular model (NSC-34, originally produced by fusing MN-rich mouse embryonic spinal cord cells with neuroblastoma N18TG2 cells as undifferentiated cells similar to NSCs) induced many changes in mitochondrial properties [149]. In the initial *Smn* knockdown phase, mitochondrial activity increased to compensate for the energy deficit, suggesting this energy demand can be overcome in early disease phases. Nevertheless, when energy demand increases, such as during MN maturation [149], SMN protein levels are insufficient. Indeed, mitochondria from iPSCs-derived MNs from SMA1 patients showed reduced trafficking, number, membrane potential, and size [154].

Therefore, it is possible that NSCs obtained from neurogenic brain areas of SMA animal models or humans or produced from iPSCs behave similarly to healthy NSCs since energy requirements at this development stage can be met without mitochondria.

## 4. Conclusions

SMA is a devastating disease affecting the young. Successful new treatments have recently been developed that can preserve MNs from a very early stage. Nevertheless, these are ineffective in patients who have already lost most of their MNs. Unfortunately, this means that many SMA1 and SMA2 patients are left behind and cannot be effectively treated with these drugs. Therefore, new interventions are required, for example, stem cell transplantation, that not only can exert exogenous replacement of lost MNs but can also have a trophic effect promoting the ability of the neurons to form new branches, innervating muscles that were uninnervated before degeneration began, or change the gene expression pattern or the endogenous MNs to the healthy phenotype [155]. 

NSCs transplantation could exert a higher impact if made using endogenous cells from the patient. In this context, the production of iPSCs from an SMA patient [1,156] and their correction using CRISPR/Cpf1 [157] or by oligodeoxynucleotide transfection [158] have a pivotal impact. The differentiation of these corrected iPSCs in healthy NSCs or MNs could allow effective results as those reported by different authors in animal models [158]. Consequently, we focused our work on the knowledge of neural stem cell physiological behavior in neurogenic areas of our CNS that could represent a possible resource to counteract or compensate for disease effects.

*SMN1*’s absence interferes with the normal NSC homeostasis but especially affects their maturation into specialized adult cells. Numerous pathways are disrupted, with implications for neighboring pathways and compromising many physiological processes. Therefore, it is important to continue advancing our understanding of the physiological characteristics disrupted by *SMN1*’s absence.

## Figures and Tables

**Figure 1 ijms-23-15209-f001:**
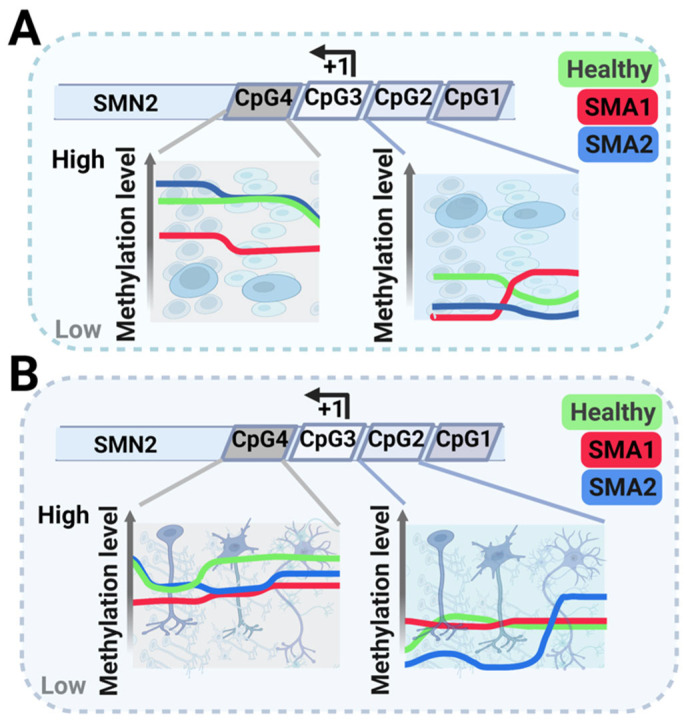
Schematic representation of methylation levels in the survival of motor neuron 2 (*SMN2*) gene at various differentiation stages during motor neuron (MN) development: induced pluripotent stem cells (iPSCs), neuroepithelial precursors (NEP) in panel (**A**), immature MNs (IMN), and mature MNs (MMNs) in panel (**B**). Green represents healthy individuals, blue represents SMA2 patients, and red represents SMA1 patients. The scheme also describes methylation in the *SMN2* gene’s four different CpG islands.

**Figure 2 ijms-23-15209-f002:**
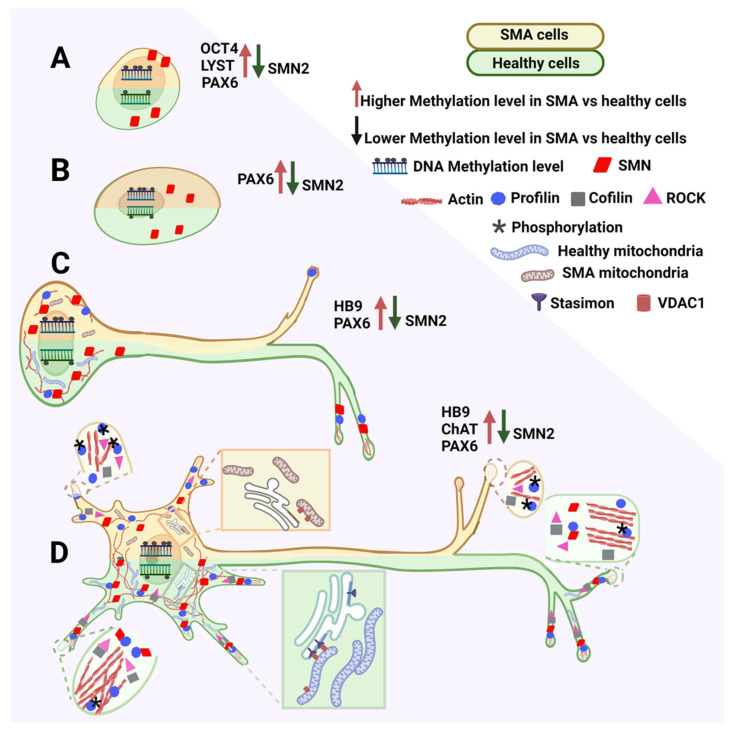
Scheme describing altered interactions in spinal muscular atrophy (SMA) cells during MN development. Each cell drawing is divided in two: the green-shaded half represents cells from healthy individuals, while the orange-shaded half represents cells from SMA patients. Reduced *SMA2* methylation is accompanied by an increase in the methylation of various genes. The low SMN protein level alters different pathways and changes protein interactions either at the soma or neurite level. SMA patient mitochondria do not interact properly with the endoplasmic reticulum and have an inappropriate morphology. (**A**): iPSCs; (**B**): NEPs; (**C**): IMNs; and (**D**): MMNs. OCT4: POU class 5 homeobox 1; LYST: lysosomal trafficking regulator; PAX6: paired box 6; HB9: motor neuron and pancreas homeobox 1 gene; and CHAT: Choline acetyltransferase.

## Data Availability

Not applicable.

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
