# Peer review of "NSC Physiological Features in Spinal Muscular Atrophy: SMN Deficiency Effects on Neurogenesis"

_ijms, 2022, doi:10.3390/ijms232315209_

Round 1
Reviewer 1 Report
In the manuscript entitled “NSC Physiological Features in Spinal Muscular Atrophy: SMN Deficiency Effects on Neurogenesis”, authors have summarized the role of low level of SMN on the physiological properties of NSCs from animals and human cellular models. This review is concise and systematically organized with relevant information under appropriate subheadings. The importance of paper is further enhanced by schematic depiction of how low level of SMN alters different pathways in spinal muscular atrophy cells during motor neuron development. However, it can be improved after addressing following points.
1. In the section “1 Neural stem cell” authors have stated many sentences without references, viz, “In recent decades, the subventricular zone (SVZ) and hippocampal dentate gyrus have been described as the two main neurogenic niches, especially in rodents. Authors should cite the original research article.
2. Cite original research article for the sentence “This functional dichotomy is an interesting aspect: while radial glial NSCs found in the adult neurogenic zones can form only new neurons in vivo, they retain the ability to produce glia in vitro”.
3. Replace ref-95 with original research article. This ref is not specific to SMA so it has many aberrations like it articulates “SMN2” as “SMA2” at many places in the relevant paragraph.
4. None of these two refs (96,97) justify the statements mentioned as “OCT4 is commonly hypomethylated in iPSCs or ESCs. However, it showed greater methylation in iPSCs from SMA1 and SMA2 patients than from healthy individuals. In contrast, the lysosomal trafficking regulator gene is hypermethylated in these cells [97] and iPSCs from SMA1 patients and healthy individuals. Therefore, author should insert suitable refs.
Plagiarism percentage - not checked by the reviewer.
Author Response
Dear Professor Reviewer 1,
First, let me express our gratitude for your helpful suggestions and your nice judgments.
Please find enclosed the revised version of our paper entitled “NSC physiological features in spinal muscular atrophy: SMN deficiency effects on neurogenesis” by Raffaella Adami and myself.
Here, we reply point-by-point to all the issues raised by the Reviewers, the manuscript was marked up using the “Track Changes” function, and in the rebuttal letter is indicated in blue.
In this file, the references are reported as first authors and year to help the reviewers.
Line numbers correspond to the new version of the paper.
In the manuscript entitled “NSC Physiological Features in Spinal Muscular Atrophy: SMN Deficiency Effects on Neurogenesis”, authors have summarized the role of low level of SMN on the physiological properties of NSCs from animals and human cellular models. This review is concise and systematically organized with relevant information under appropriate subheadings. The importance of paper is further enhanced by schematic depiction of how low level of SMN alters different pathways in spinal muscular atrophy cells during motor neuron development. However, it can be improved after addressing following points.
- In the section “1 Neural stem cell” authors have stated many sentences without references, viz, “In recent decades, the subventricular zone (SVZ) and hippocampal dentate gyrus have been described as the two main neurogenic niches, especially in rodents. Authors should cite the original research article.
Thanks to the Reviewer 1 for marking this point we introduced the original article as suggested, moreover, we added other original article references in appropriate points of section 1.
Line 53
The exceptions are radial glial NSCs found in adult neurogenic zones (niches). In recent decades, the subventricular zone (SVZ) (Reynolds and Weiss, 1992; Lois and Alvarez-Buylla, 1993) and hippocampal dentate gyrus (Altman and Das, 1965) have been described as the two main neurogenic niches, especially in rodents.
Line 56
However, it is possible to identify and isolate this cell type in other brain regions in some animals (Pagano et al., 2000; Tropepe et al., 2000).
Line 59
For example, the rodent SVZ has a homeostatic function, producing cells that replace glomerular and periglomerular neurons in their olfactory bulb (Rousselot et al., 1995).
Line 79
The first NSC feature is their ability to self-maintain. Each NSC performs self-maintenance either through forced asymmetric division, where they generate another stem cell and a cell destined to differentiate, or stochastic differentiation, where a stem cell population is conserved by approximately equal numbers of stem cells generating other stem cells via self-replication and generating cells destined to differentiate (Morshead et al., 1998; Obernier et al., 2018).
- Cite original research article for the sentence “This functional dichotomy is an interesting aspect: while radial glial NSCs found in the adult neurogenic zones can form only new neurons in vivo, they retain the ability to produce glia in vitro”.
Line 89
This functional dichotomy is an interesting aspect: while radial glial NSCs found in the adult neurogenic zones can form only new neurons in vivo, they retain the ability to produce glia in vitro because the exposure to epidermal growth factor arrests the production of neuroblasts becoming more astrocytes like (Doetsch et al., 2002).
- Replace ref-95 with original research article. This ref is not specific to SMA so it has many aberrations like it articulates “SMN2” as “SMA2” at many places in the relevant paragraph.
We completely agreed with Reviewer 1 as the Hauke paper mix up SMN2 with SMA2.
Line 287
Fibroblast cell lines from SMA patients showed almost complete methylation at CpG1 and CpG4, while CpG2 was partially methylated and CpG3 showed very little methylation (Hauke et al., 2009). CpG methylation (at positions -290 and -296) in blood-borne cells correlated with disease severity and first transcriptional start site activity at SMN2 at position -296 in patients affected by severe SMA and suffering from mild SMA but carrying identical SMN2 copy numbers (Hauke et al., 2009). Inhibiting SMN2 silencing via DNA methylation is considered a promising pharmacologic SMA therapy (Hauke et al., 2009).
- None of these two refs (96,97) justify the statements mentioned as “OCT4 is commonly hypomethylated in iPSCs or ESCs. However, it showed greater methylation in iPSCs from SMA1 and SMA2 patients than from healthy individuals. In contrast, the lysosomal trafficking regulator gene is hypermethylated in these cells [97] and iPSCs from SMA1 patients and healthy individuals. Therefore, author should insert suitable refs.
Thanks to the Reviewer 1 for underlying this point we introduced the articles suitable to clarify these points.
Line 315
The same study (Maretina et al., 2022) compared the DNA methylation of genes involved in differentiation from iPSCs to NEP to MNs, starting from genes involved in maintaining the pluripotency state particularly relevant to iPSCs. OCT4 is commonly hypomethylated in ESCs or iPSCs (Hattori et al., 2004; Bhutani et al., 2010). However, it showed greater methylation in iPSCs from SMA1 and SMA2 patients than from healthy individuals. In contrast, the lysosomal trafficking regulator gene is hypermethylated in these cells (Nishino et al., 2011) and iPSCs from SMA1 patients and healthy individuals
Plagiarism percentage - not checked by the reviewer.
We hope that the revised version of the paper will fulfill the requirements for publication in the International Journal of Molecular Science.
Please have the most cordial regards, from my collaborator and myself.
Yours sincerely,
Daniele Bottai, Ph. D.
daniele.bottai@unimi.it
University of Milan
Assistant Professor in Physiology
Department of Pharmaceutical Sciences
Section of Pharmacology and Biosciences
University of Milan
Via Balzaretti 9, 20133 Milan ITALY
Tel 0250318233
References
Altman J, Das GD (1965) Autoradiographic and histological evidence of postnatal hippocampal neurogenesis in rats. J Comp Neurol 124:319-335.
Bhutani N, Brady JJ, Damian M, Sacco A, Corbel SY, Blau HM (2010) Reprogramming towards pluripotency requires AID-dependent DNA demethylation. Nature 463:1042-1047.
Doetsch F, Petreanu L, Caille I, Garcia-Verdugo JM, Alvarez-Buylla A (2002) EGF converts transit-amplifying neurogenic precursors in the adult brain into multipotent stem cells. Neuron 36:1021-1034.
Hattori N, Nishino K, Ko YG, Ohgane J, Tanaka S, Shiota K (2004) Epigenetic control of mouse Oct-4 gene expression in embryonic stem cells and trophoblast stem cells. J Biol Chem 279:17063-17069.
Hauke J, Riessland M, Lunke S, Eyüpoglu IY, Blümcke I, El-Osta A, Wirth B, Hahnen E (2009) Survival motor neuron gene 2 silencing by DNA methylation correlates with spinal muscular atrophy disease severity and can be bypassed by histone deacetylase inhibition. Hum Mol Genet 18:304-317.
Lois C, Alvarez-Buylla A (1993) Proliferating subventricular zone cells in the adult mammalian forebrain can differentiate into neurons and glia. Proc Natl Acad Sci U S A 90:2074-2077.
Maretina MA, Valetdinova KR, Tsyganova NA, Egorova AA, Ovechkina VS, Schiöth HB, Zakian SM, Baranov VS, Kiselev AV (2022) Identification of specific gene methylation patterns during motor neuron differentiation from spinal muscular atrophy patient-derived iPSC. Gene 811:146109.
Morshead CM, Craig CG, van der Kooy D (1998) In vivo clonal analyses reveal the properties of endogenous neural stem cell proliferation in the adult mammalian forebrain. Development 125:2251-2261.
Nishino K, Toyoda M, Yamazaki-Inoue M, Fukawatase Y, Chikazawa E, Sakaguchi H, Akutsu H, Umezawa A (2011) DNA methylation dynamics in human induced pluripotent stem cells over time. PLoS Genet 7:e1002085.
Obernier K, Cebrian-Silla A, Thomson M, Parraguez JI, Anderson R, Guinto C, Rodas Rodriguez J, Garcia-Verdugo JM, Alvarez-Buylla A (2018) Adult Neurogenesis Is Sustained by Symmetric Self-Renewal and Differentiation. Cell Stem Cell 22:221-234.e228.
Pagano SF, Impagnatiello F, Girelli M, Cova L, Grioni E, Onofri M, Cavallaro M, Etteri S, Vitello F, Giombini S, Solero CL, Parati EA (2000) Isolation and characterization of neural stem cells from the adult human olfactory bulb. Stem Cells 18:295-300.
Reynolds BA, Weiss S (1992) Generation of neurons and astrocytes from isolated cells of the adult mammalian central nervous system. Science 255:1707-1710.
Rousselot P, Lois C, Alvarez-Buylla A (1995) Embryonic (PSA) N-CAM reveals chains of migrating neuroblasts between the lateral ventricle and the olfactory bulb of adult mice. J Comp Neurol 351:51-61.
Tropepe V, Coles BL, Chiasson BJ, Horsford DJ, Elia AJ, McInnes RR, van der Kooy D (2000) Retinal stem cells in the adult mammalian eye. Science 287:2032-2036.
Reviewer 2 Report
The manuscript entitled "NSC physiological features in spinal muscular atrophy: SMN deficiency effects on neurogenesis" reviewed the characteristics of neural stem cells collected from patients with spinal muscular atrophy (SMA). As the authors mention, due to new therapeutic drugs, interest in SMA, the therapy, and the therapy's effectiveness seems to be increasing recently. Otherwise, the review needs significant improvements.
Major comments
1. In both the abstract and conclusion, the authors refer to stem cells as therapeutic targets for SMA1 and SMA2 patients who have lost motor neurons. I think the authors should also mention stem cell transplantation when talking about the possibility of stem cell therapy, such as the classic paper by Corti et al. [1] on stem cell transplantation. However, the authors did not mention stem cell transplantation at all in the manuscript.
[1] Corti S, Nizzardo M, Nardini M, et al. Neural stem cell transplantation can ameliorate the phenotype of a mouse model of spinal muscular atrophy. J Clin Invest. 2008;118:3316-30. doi: 10.1172/JCI35432.
2. Although the authors well document the properties of SMA stem cells, they do not address any studies that have examined interventions on SMA stem cells. Do the authors believe that SMA stem cells themselves can be used to treat SMA?
3. If this paper deals mainly with the properties of SMA stem cells, it should be said that this paper is concerned with the molecular pathology and neuronal circuit formation disorders of SMA fetuses, and not directly related to treatment of SMA. Therefore, this article is a review of the pathogenesis of SMA, not its treatment.
4. Based on the points mentioned above, the authors should revise the wording of the abstract and conclusion.
5. The authors identify eight clinical types of SMA but describe only seven types. In addition, the paper that the authors used as the basis for their claim that "there are eight clinical types of SMA" is their own previous paper, and the original source [2] is not cited. Furthermore, the notation is different from the original paper (3a and 3b in the original, 3A and 3B in this paper).
[2] Talbot K, Tizzano EF. The clinical landscape for SMA in a new therapeutic era. Gene Ther. 2017;24:529-533. doi: 10.1038/gt.2017.52.
Minor comments
Line 5; The authors' institutions seem to be same, though institution numbers, which were placed by the authors’ names, were different. Please check that this is correct.
Line 85; There does not appear to be a subsection "1.1".
Line 106; "2. . Spinal muscular atrophy" should be "2. Spinal muscular atrophy".
Line 208; "3. . Physiological and molecular differences in healthy and SMA NSCs" should be "3. Physiological and molecular differences in healthy and SMA NSCs".
Line 466; "4 . Conclusions" should be "4. Conclusions".
Author Response
Dear Professor Reviewer 2,
First, let me express our gratitude for your helpful suggestions and your nice judgments.
Please find enclosed the revised version of our paper entitled “NSC physiological features in spinal muscular atrophy: SMN deficiency effects on neurogenesis” by Raffaella Adami and myself.
Here, we reply point-by-point to all the issues raised by the Reviewers, the manuscript was marked up using the “Track Changes” function, and in the rebuttal letter is indicated in blue.
In this file, the references are reported as first authors and year to help the reviewers.
Line numbers correspond to the new version of the paper.
The manuscript entitled "NSC physiological features in spinal muscular atrophy: SMN deficiency effects on neurogenesis" reviewed the characteristics of neural stem cells collected from patients with spinal muscular atrophy (SMA). As the authors mention, due to new therapeutic drugs, interest in SMA, the therapy, and the therapy's effectiveness seems to be increasing recently. Otherwise, the review needs significant improvements.
Major comments
- In both the abstract and conclusion, the authors refer to stem cells as therapeutic targets for SMA1 and SMA2 patients who have lost motor neurons. I think the authors should also mention stem cell transplantation when talking about the possibility of stem cell therapy, such as the classic paper by Corti et al. [1] on stem cell transplantation. However, the authors did not mention stem cell transplantation at all in the manuscript.
[1] Corti S, Nizzardo M, Nardini M, et al. Neural stem cell transplantation can ameliorate the phenotype of a mouse model of spinal muscular atrophy. J Clin Invest. 2008;118:3316-30. doi: 10.1172/JCI35432.
We thank the Reviewer 2 for mentioning this point. We made the appropriate changes.
Line 514
Therefore, new interventions are required, for example, stem cell transplantation, that not only can exert exogenous replacement of lost MNs but can have also a trophic effect promoting the ability of the neurons to form new branches, innervating muscles that were uninnervated before degeneration began, or change the gene expression pattern or the endogenous MNs to the healthy phenotype (Corti et al., 2008).
- Although the authors well document the properties of SMA stem cells, they do not address any studies that have examined interventions on SMA stem cells. Do the authors believe that SMA stem cells themselves can be used to treat SMA?
We appreciated the Reviewer 2 suggestion. Indeed, in the new version, we discussed the importance of the correction of SMA cells (iPSCs) and their use in more differentiated forms such as NSCs or MNs.
Line 520
NSCs transplantation could exert a higher impact if made using endogenous cells from the patient. In this context, the production of iPSCs from an SMA patient (Adami and Bottai, 2019; Valetdinova et al., 2019) and their correction using CRISPR/Cpf1 (Zhou et al., 2018) or by oligodeoxynucleotide transfection (Corti et al., 2012) have a pivotal impact. The differentiation of these corrected iPSCs in healthy NSCs or motor neurons could allow effective results as those reported by different authors in animal models (Corti et al., 2012).
- If this paper deals mainly with the properties of SMA stem cells, it should be said that this paper is concerned with the molecular pathology and neuronal circuit formation disorders of SMA fetuses, and not directly related to treatment of SMA. Therefore, this article is a review of the pathogenesis of SMA, not its treatment.
We changed the abstract and the conclusion accordingly to the Reviewer 2 to support these suggested points, as shown in point 4.
- Based on the points mentioned above, the authors should revise the wording of the abstract and conclusion.
Abstract
Line 20
This review summarizes current knowledge on the physiological properties of NSCs from animals and human cellular models with an SMA background converging on the molecular and neuronal circuit formation alterations of SMA fetuses and it is not focused on the treatment of SMA. By understanding how SMA alters NSC physiology, we can identify new and promising interventions that could help support affected patients.
Conclusion
Line 514
Therefore, new interventions are required, for example, stem cell transplantation that not only can exert exogenous replacement of lost MNs but can have also a trophic effect promoting the ability of the neurons to form new branches, innervating muscles that were uninnervated before degeneration began, or change the gene expression pattern or the endogenous MNs to the healthy phenotype (Corti et al., 2008).
NSCs transplantation could exert a higher impact if made using endogenous cells from the patient. In this context, the production of iPSCs from an SMA patient (Adami and Bottai, 2019; Valetdinova et al., 2019) and their correction using CRISPR/Cpf1 (Zhou et al., 2018) or by oligodeoxynucleotide transfection (Corti et al., 2012) have a pivotal impact. The differentiation of these corrected iPSCs in healthy NSCs or MNs could allow effective results as those reported by different authors in animal models (Corti et al., 2012). Consequently, we focused our work on the knowledge of neural stem cell physiological behavior in neurogenic areas of our CNS that could represent a possible resource to counteract or compensate for disease effects.
- The authors identify eight clinical types of SMA but describe only seven types. In addition, the paper that the authors used as the basis for their claim that "there are eight clinical types of SMA" is their own previous paper, and the original source [2] is not cited. Furthermore, the notation is different from the original paper (3a and 3b in the original, 3A and 3B in this paper).
[2] Talbot K, Tizzano EF. The clinical landscape for SMA in a new therapeutic era. Gene Ther. 2017;24:529-533. doi: 10.1038/gt.2017.52.
We thank the Reviewer 2 for making this point, we corrected the text as indicated.
The original Article (Talbot and Tizzano, 2017) was introduced, and the notation was changed.
Line 126
It displays a wide clinical spectrum, from embryonic lethality to adult-onset, and can be divided into eight different phenotypes (Talbot and Tizzano, 2017; Adami and Bottai, 2019). Type 0, also called 1A, is diagnosed during prenatal life since the fetus reduces its movements; soon after birth, patients readily require respiratory support and die within the first month. Types 1B and 1C, also called Werdnig–Hoffman disease, are also severe SMA forms. Their onset is before 6 months and is indicated by the patient’s inability to sit, so they remain prone and require respiratory support; if untreated, patients die early, but with respiratory support, they can live for two years on average. Type 2 is an intermediate phenotype with onset between 7 and 18 months; patients can sit but cannot walk and usually live for >40 years. Types 3 and 4 are less severe forms. Type 3 has an onset after 18 months of life and can be divided into 3a, where the patient loses the ability to walk during adulthood, and 3b, where the patient retains the ability to walk; their life span is similar to the general population. Type 4 is so mild that, in many cases, it cannot be diagnosed; its onset is in the second and third decades of life, and patients have a similar walking capacity and life span to the general population (Mercuri et al., 2012; Finkel et al., 2014; Talbot and Tizzano, 2017; Adami and Bottai, 2019; López-Cortés et al., 2022).
Minor comments
Line 5; The authors' institutions seem to be same, though institution numbers, which were placed by the authors’ names, were different. Please check that this is correct.
We corrected this point by putting the same number to both authors.
Line 4
Raffaella Adami1 and Daniele Bottai1*
1 Department of Pharmaceutical Sciences, Section of Pharmacology and Biosciences, University of Milan, Via Balzaretti 9, 20133 Milan, ITALY.
* Correspondence: daniele.bottai@unimi.it; Tel.: +39 050318233
Line 85; There does not appear to be a subsection "1.1".
This subsection number was corrected:
Line 94
1.1 Induced pluripotent stem cells (iPSCs) as NSC precursors
Line 106; "2. . Spinal muscular atrophy" should be "2. Spinal muscular atrophy".
We made this correction:
Line 117
- Spinal muscular atrophy
Line 208; "3. . Physiological and molecular differences in healthy and SMA NSCs" should be "3. Physiological and molecular differences in healthy and SMA NSCs".
We made this correction
Line 227
- Physiological and molecular differences in healthy and SMA NSCs
Line 466; "4 . Conclusions" should be "4. Conclusions".
We made this correction
Line 509
- Conclusions
We hope that the revised version of the paper will fulfill the requirements for publication in the International Journal of Molecular Science.
Please have the most cordial regards, from my collaborator and myself.
Yours sincerely,
Daniele Bottai, Ph. D.
daniele.bottai@unimi.it
University of Milan
Assistant Professor in Physiology
Department of Pharmaceutical Sciences
Section of Pharmacology and Biosciences
University of Milan
Via Balzaretti 9, 20133 Milan ITALY
Tel 0250318233
References
Adami R, Bottai D (2019) Spinal Muscular Atrophy Modeling and Treatment Advances by Induced Pluripotent Stem Cells Studies. Stem Cell Rev Rep 15:795-813.
Corti S, Nizzardo M, Simone C, Falcone M, Nardini M, Ronchi D, Donadoni C, Salani S, Riboldi G, Magri F, Menozzi G, Bonaglia C, Rizzo F, Bresolin N, Comi GP (2012) Genetic correction of human induced pluripotent stem cells from patients with spinal muscular atrophy. Sci Transl Med 4:165ra162.
Corti S, Nizzardo M, Nardini M, Donadoni C, Salani S, Ronchi D, Saladino F, Bordoni A, Fortunato F, Del Bo R, Papadimitriou D, Locatelli F, Menozzi G, Strazzer S, Bresolin N, Comi GP (2008) Neural stem cell transplantation can ameliorate the phenotype of a mouse model of spinal muscular atrophy. J Clin Invest 118:3316-3330.
Finkel RS et al. (2014) Observational study of spinal muscular atrophy type I and implications for clinical trials. Neurology 83:810-817.
López-Cortés A, Echeverría-Garcés G, Ramos-Medina MJ (2022) Molecular Pathogenesis and New Therapeutic Dimensions for Spinal Muscular Atrophy. Biology (Basel) 11.
Mercuri E, Bertini E, Iannaccone ST (2012) Childhood spinal muscular atrophy: controversies and challenges. Lancet Neurol 11:443-452.
Talbot K, Tizzano EF (2017) The clinical landscape for SMA in a new therapeutic era. Gene Ther 24:529-533.
Valetdinova KR, Maretina MA, Kuranova ML, Grigor'eva EV, Minina YM, Kizilova EA, Kiselev AV, Medvedev SP, Baranov VS, Zakian SM (2019) Generation of two spinal muscular atrophy (SMA) type I patient-derived induced pluripotent stem cell (iPSC) lines and two SMA type II patient-derived iPSC lines. Stem Cell Res 34:101376.
Zhou M, Hu Z, Qiu L, Zhou T, Feng M, Hu Q, Zeng B, Li Z, Sun Q, Wu Y, Liu X, Wu L, Liang D (2018) Seamless Genetic Conversion of SMN2 to SMN1 via CRISPR/Cpf1 and Single-Stranded Oligodeoxynucleotides in Spinal Muscular Atrophy Patient-Specific Induced Pluripotent Stem Cells. Hum Gene Ther 29:1252-1263.
Round 2
Reviewer 2 Report
Generally, I am satisfied with the revisions that have been made by the authors. Although the authors responded well to my comments, several minor changes may still be required.
Lines 91-92
…because the exposure to epidermal growth factor arrests the production of neuroblasts becoming more astrocytes like [25].
If you want to make an adjective here, it should be “astrocyte(s)-like”. Please use hyphen.
Lines 523-525
The differentiation of these corrected iPSCs in healthy NSCs or MNs could allow effective results as those reported by different authors in animal models [157,158].
When you say “different authors”, you should cite at least two references. In the revised version, you cited only one reference, [158] (Corti et al., 2012). I think you should also cite [157] (Zhou et al., 2018).